# Mycobacterial heparin-binding hemagglutinin (HBHA)-induced interferon-γ release assay (IGRA) for discrimination of latent and active tuberculosis: A systematic review and meta-analysis

**Jinhua Tang**, **Yuan Huang**, **Zheng Cai**, **Yueyun Ma**\*

Department of Clinical Laboratory, Air Force Medical Centre, Air Force Medical University, Beijing, China

\* mayueyun2020@163.com

## Abstract

### Background

The Mycobacterial heparin-binding hemagglutinin (HBHA) is an important latency-associated antigen that can be used to distinguish between latent tuberculosis infection (LTBI) and active tuberculosis (ATB). Although many studies were explored the efficiency of the HBHA-induced interferon-γ release assay (IGRA) in different populations, the clinical differential value of HBHA-IGRA is still controversial. Therefore, the aim of this study was to determine whether the HBHA-IGRA can be used as an efficient test for the discrimination of LTBI and ATB by a systematic review and meta-analysis.

### Methods

Relevant articles were retrieved from PubMed, Embase, Web of Science, and the Cochrane Library on Oct 18, 2020, with no start date limitation. The quality of each study was evaluated using Review Manager 5.4. The Stata MP v.14.0 software was used to combine sensitivity, specificity, likelihood ratio (LR), diagnostic odds ratio (DOR), summary receiver operating characteristic (SROC) curve, and area under SROC (AUC) to evaluate the diagnostic value of HBHA-IGRA for discrimination of LTBI and ATB. Meta-regression and subgroup analysis were performed for the sources of heterogeneity based on the selection criteria for active TB, the population, the TB burden, the type of antigen, the type of sample, and the time of antigen stimulation.

### Results

A total of 13 studies (14 results) were included in this meta-analysis, including 603 ATB patients and 514 LTBI individuals. The pooled sensitivity and specificity of the HBHA-IGRA for discrimination of the LTBI and ATB were 0.70 (95% CI, 0.57~0.80) and 0.78 (95% CI, 0.71~0.84), respectively. The pooled positive likelihood ratio (PLR), negative likelihood ratio (NLR), and diagnostic odds ratio (DOR) were 3.15 (95%CI, 2.43~4.09), 0.39 (95% CI,

**Data Availability Statement:** All relevant data are within the manuscript and its Supporting Information files.

**Funding:** This work was supported by a grant from the National Natural Science Foundation of China (81371857).

**Competing interests:** The authors have declared that no competing interests exist.

0.27~0.56), and 8.11 (95% CI, 4.81~13.67), respectively. The AUC was 0.81 (95% CI, 0.77~0.84). The subgroup analysis showed that the main source of heterogeneity was due to the HIV-infected population incorporated, and the different selection criteria of active TB subjects would also lead to the variation of the pooled sensitivity and specificity. Different TB burdens, HBHA antigen types, sample types, antigen stimulation time and BCG vaccination did not affect the heterogeneity in this analysis.

## Conclusion

The HBHA-IGRA is a promising immunodiagnostic test for discrimination of latent and active TB, which can be added in commercial IGRAs to enhance the differential diagnostic performance.

## 1. Introduction

Tuberculosis (TB) is one of the infectious diseases causing high morbidity and mortality worldwide and remains an important global public health concern. The WHO global tuberculosis report [1] depicts that about a quarter of the world's population has been infected with *Mycobacterium tuberculosis* (*Mtb*). Only about 5–10% of the infected population develops active TB while many of them have asymptomatic "latent tuberculosis infection (LTBI)". Although many of the LTBIs are not infectious and do not produce active disease, some of the latent infections can become active infections, especially in people with a weak immune system. Therefore, the surveillance and management of latent tuberculosis infections are also critically important to greatly reduce the global burden of TB [2].

The early identification of LTBI and active TB is critical in reducing the global burden of TB. The development of a latent infection or an active case after the entry of *Mtb* into the body depends on a variety of factors including the most important immune status. The WHO recommends that an interferon (IFN)-γ release assay (IGRA) or a tuberculin skin test (TST) can be used to screen for TB infections [3]. The IGRA which measures IFN-γ secretion stimulated by the *Mtb*-specific antigens such as ESAT-6 and CFP-10 has a better predictive value than the traditional TST which is based on the purified protein derivative (PPD) in projecting the TB progression [4]. However, neither the IGRAs nor the TST is useful in discriminating the LTBI and the active TB [5, 6]. Therefore, there is currently no efficient test to directly identify the status of *Mtb* infection in humans.

The mycobacterial heparin-binding hemagglutinin (HBHA) is a major latency antigen associated with the dormancy of the *Mtb* and LTBI [7, 8]. Many studies [9–11] showed that the HBHA has a discriminatory potential in differentiating the latent and active TB, especially with the release of interferon- γ. There have been many reports of using the HBHA-based IGRAs until now. However, due to the lack of a large sample size and the controversies among different results in previous studies, the clinical use of the HBHA-IGRA to differentiate active TB from the LTBI has not been popularized.

To summarize the current state of the research and evaluate the diagnostic value of the HBHA-IGRA, we performed a systematic review and meta-analysis on previous human studies that used the HBHA as a stimulating antigen in the IGRA tests for the diagnosis of active TB and LTBI. The aim of this study was to identify the efficacy of the HBHA-IGRA as a good differential diagnostic method for active and latent *Mtb* infection and to provide a basis for its clinical utilization.

## 2. Materials and methods

The systematic review and the meta-analysis in this study, were conducted strictly following the criteria of the Preferred Reporting Items for Systematic Reviews and Meta-Analyses (PRISMA) statement.

### 2.1. Search strategy

In this systematic review and meta-analysis, the PubMed, Embase, Web of Science (WOS), and the Cochrane Library databases were searched for the relevant studies in English on Oct 18, 2020, with no start date limitation. The search terms were as follows: ("tuberculosis" OR "tuberculous" OR "tubercular" OR "TB" OR "mycobacterium" OR "mycobacterial") AND ("interferon-gamma" OR "gamma interferon" OR "IFN gamma" OR "Interferon-γ" OR "IFN-γ" OR "interferon gamma release assays" OR "Interferon-gamma Release Test" OR "IGRA" OR "T cell assay" OR "T cell response" OR "enzyme-linked immunospot" OR "ELISpot") AND ("heparin-binding hemagglutinin adhesin" OR "heparin-binding hemagglutinin" OR "HBHA"). Additionally, we manually searched the reference list of related articles for the other potentially relevant studies.

### 2.2. Study selection criteria

All relevant studies included in the meta-analysis must meet the following criteria: (1) the study had the discrimination analysis of the latent tuberculosis infection (LTBI) and active tuberculosis (ATB), (2) subjects in the study included both individuals with the LTBI and patients with the ATB, (3) using mycobacterial heparin-binding hemagglutinin (HBHA) as stimulating antigen and indicators to be evaluated including IFN-γ, (4) studies with a clear diagnostic cut-off value or studies directly or indirectly extracted the true positive (TP), false positive (FP), true negative (TN) and false negative (FN) values of the HBHA-IGRA for the discrimination of LTBI and ATB to construct a diagnostic four-grid table.

The LTBI group was defined as individuals who were selected based on the positive tuberculin skin test (TST) (HIV-uninfected people≥10 mm induration and HIV-infected people≥5 mm induration) or the IGRA tests which had no signs or symptoms of active TB but were at risk for the active TB disease based on the WHO's recommendation [2]. The ATB group was defined as patients with microbiologically confirmed TB or high clinical suspicion and a positive response to anti-TB treatment, who were untreated or treated within four weeks. The individuals who were infected with the human immunodeficiency virus (HIV) were also included in this study.

The reviews, letters, abstracts, case reports, duplicated studies, studies that did not include the integrated date, studies written in languages other than English, and studies that did not involve humans were excluded from this meta-analysis.

### 2.3. Data extraction and quality assessment

To compute this systematic review and meta-analysis, two authors independently conducted the data extraction. The quality of the literature was also evaluated by these two authors based on the inclusion and exclusion criteria to include in the meta-analysis. The disagreements between these two individuals' evaluations were resolved by consensuses. For each study, the basic information and relevant results including the first author and the year of publication; the time when the study performed; country; study design; population; the proportion of individuals who were BCG vaccinated; the total number of cases enrolled; age characteristics; the number of males/females; the number of active TB/LTBI; the definition of active TB; clinical

subtype of active TB; antigen type; assay type; stimulation time; cut-off value and sample type in the studies included, were collected. The quality of each study was assessed by the Quality Assessment of Diagnostic Accuracy Studies (QUADAS-2). The risk of bias included four parts: Patient selection, Index test, Reference standard, and Flow and timing. The studies with a high risk of bias were determined as poor quality and those with low risk as good quality. The results of the quality assessment were summarized and graphed using Review Manager (RevMan) v.5.4.1. (The Cochrane Collaboration, 2020.)

### 2.4. Statistical analysis

The meta-analysis was performed using the Stata MP v.14.0 software (StataCorp, LLC, College Station, TX, USA). The pooled sensitivity and specificity, pooled positive likelihood ratio (+LR) and negative likelihood ratio (-LR), diagnostic odds ratio (DOR), and summary receiver operating characteristic (SROC) curve were computed. The heterogeneity caused by the threshold effect was examined by the Spearman correlation analysis. The heterogeneities of sensitivity, specificity, +LR, -LR, and DOR were assessed by the Higgins $I^2$ statistic and Cochran's Q test. If an $I^2$ value was>50%, it suggested a significant heterogeneity, and the meta-regression and the subgroup analysis were used to identify the source of the heterogeneity [12]. The Deeks' funnel plot asymmetry test was used to assess the publication bias and the $p<0.05$ was considered statistically significant.

## 3. Results

### 3.1. Search results and the study characteristics

A total of 168 relevant studies were retrieved from three independent online databases, and 92 duplicated elucidations were removed from the further analysis. Subsequently, after reviewing the title and abstract of these elucidations, only 30 articles directly related to the objective continued to remain. Among these 30 articles, eight articles had no data on patients with ATB or individuals with LTBI, three studies did not mention the use of the IGRA test, five elucidations did not provide sufficient data for meta-analysis, and one study lacked the cut-off value to construct a diagnostic four-grid table despite having sufficient data. After all these filterings, 14 results from 13 studies were eligible for the meta-analysis and included in the current study. The details of the study screening process are shown in Fig 1.

The 13 studies (14 sets of results) were conducted from 1999 to 2019 and consisted of 603 patients with active TB and 514 individuals with LTBI. These studies were mainly performed in three countries: Italy (46%), Belgium (31%), and China (15%). The study subjects mainly were non-HIV infected (87%), from areas with low TB burden (72%), and most adults. The IFN-γ was measured mainly using enzyme-linked immunosorbent assay (ELISA) (86%), all the antigens used in the IGRAs in the studies selected were either natural or recombinant HBHA protein. All selected studies are prospective case-control studies. The detailed characteristics of all the studies are shown in Table 1. Table 2 summarizes the data extraction results from each study (2 × 2 table).

### 3.2. Quality assessment and publication bias

Pertinent to the quality assessment, the evaluations from the two independent authors were highly consistent (Fig 2). Since almost all studies (12/13) were case-control studies, the bias in patient selection was judged as "high risk" in most of the studies (8/13). The high risk of bias for the "Index Test" (7/13) largely resulted from the non-pre-specified threshold (cut-off value), and a lack of information on blind testing led to "unclear" results (4/13).

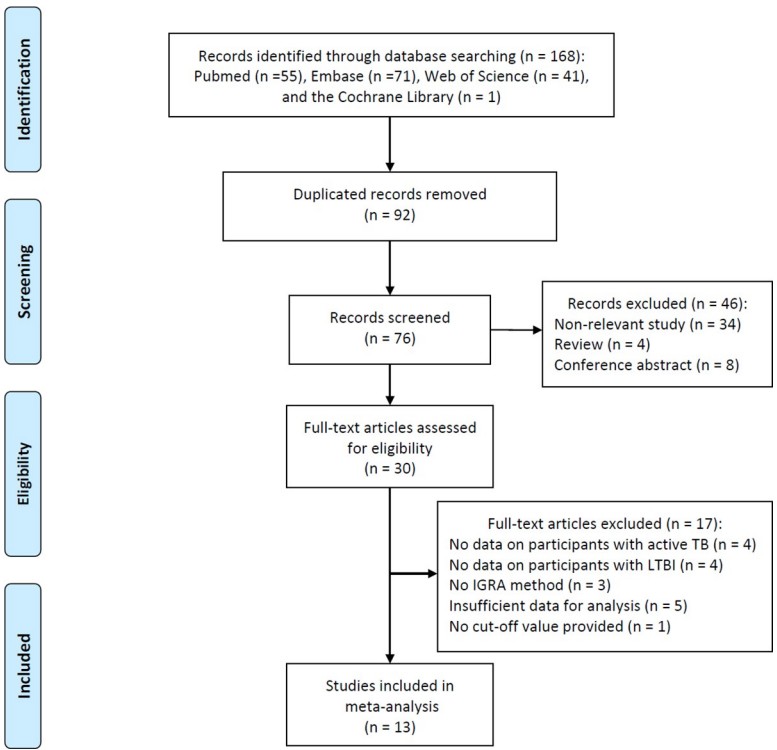

**Fig 1. Flow diagram of the study screening process.**

To evaluate the potential publication bias in these studies, the Deeks' funnel plot asymmetry test was performed (Fig 3). The *p*-value was 0.28, which indicated that no significant publication bias was found among the included studies.

### 3.3. Threshold and diagnostic accuracy of the HBHA-IGRA for discrimination of LTBI and active TB

The Spearman correlation coefficient of the 13 selected studies (14 sets of results) in the meta-analysis was 0.381 (*p* = 0.179), suggesting that there was no significant heterogeneity caused by the diagnostic threshold effect, although different cut-off values were adopted by different research teams.

The diagnostic performance results of the HBHA-IGRA tests are presented in Table 2. The pooled sensitivity and specificity of the HBHA-IGRA for discrimination of the LTBI and ATB were 0.70 (95% CI, 0.57~0.80) and 0.78 (95% CI, 0.71~0.84), respectively (Fig 4A). The pooled estimates for the positive diagnostic LR (DLR), negative DLR, diagnostic score, and DOR were 3.15 (95%CI, 2.43~4.09), 0.39 (95% CI, 0.27~0.56), 2.09 (95% CI, 1.57~2.62), and 8.11 (95% CI, 4.81~13.67), respectively (Fig 4B and 4C). The area under the SROC curve (AUC) was 0.81 (95% CI, 0.77~0.84) (Fig 5). A significant heterogeneity was observed in the above-pooled results, based on $I^2$ values of 90.7% for sensitivity, 80.1% for specificity, 64.4% for positive DLR, 90.5% for negative DLR, 68.9% for the diagnostic score, and 100.0% for DOR.

### 3.4. Meta-regression and subgroup analysis

Meta-regression and subgroup analysis were performed to explore the sources of heterogeneity in the studies that used the HBHA-IGRA test for discrimination of LTBI and ATB. Meta-

**Table 1. Characteristics of studies included in the meta-analysis.**

| | Time study performed | Country (TB burden) | Study design | Population | BCG vaccinated (%) | Participant number | Age group | Male/female | Active TB/LTBI | Active TB definition | Active TB Type | Antigen type | Assay type | Stimulation time (hour) | Cut-off Value | Sample type |
|---|---|---|---|---|---|---|---|---|---|---|---|---|---|---|---|---|
| Masungi 2002 [13] | NA | Belgium (low) | Prospective | Contact individuals and patients | NA | 49 | NA | NA | 24/25 | microbiologically confirmed TB | 81% PTB / 19% EPTB | nHBHA | IFA | 96 | 100pg/mL | PBMCs |
| Temmerman 2004 [14] | NA | Belgium (low) | Prospective | Contact individuals and patients | 0% | 101 | NA | NA | 46/55 | microbiologically confirmed TB | NA | nHBHA | ELISA | 96 | 100pg/mL | PBMCs |
| Hougardy 2007 [15] | 1999–2007 | Belgium (low) | Prospective | Students, household contacts, HCWs and patients | 40% LTBI | 149 | Adults | NA | 86/63 | clinically confirmed TB | 65% PTB / 35% EPTB | nHBHA | ELISA | 96 | 100pg/mL | PBMCs |
| Delogu 2011 [a,16] | NA | Italy (low) | Prospective | Contact individuals and patients | 37% | 87 | Adults | 53/32 | 61/26 | microbiologically confirmed TB | 100% PTB | rHBHAms | ELISA | 24 | 0.25IU/mL | Whole blood |
| Delogu 2011 [b,16] | NA | Italy (low) | Prospective | Contact individuals and patients | 37% | 72 | Adults | NA | 52/20 | microbiologically confirmed TB | 100% PTB | rHBHAms | ELISA | 168 | 0.75IU/mL | Whole blood |
| Molicotti 2011 [17] | NA | Italy (low) | Prospective | Contact individuals and patients | NA | 63 | NA | NA | 40/23 | microbiologically confirmed TB | NA | rHBHAms | ELISA | 24 | 0.25IU/mL | Whole blood |
| Wyndham-thomas 2014 [18] | NA | Belgium (low) | Prospective | Contact individuals and patients | 45% | 49 | adults | 24/25 | 17/32 | clinically confirmed TB | NA | nHBHA | ELISA | 24 | 50pg/mL | PBMCs |
| Molicotti 2015 [19] | NA | Italy (low) | Prospective | Contact individuals and patients | NA | 83 | Mainly adults | NA | 27/56 | microbiologically confirmed TB | NA | rHBHAms | ELISA | 24 | 0.20IU/mL | Whole blood |
| Wen 2017 [20] | 2016.06–2016.12 | China (high) | Prospective | Contact individuals and patients | 100% | 101 | Adults | 65/36 | 86/15 | clinically confirmed TB | 65% PTB / 35% EPTB | rHBHAms | ELISPOT | 18–20 | 6 SFCs/$10^6$ cells | PBMCs |
| Chiacchio 2017 [21] | 2012–2015 | Italy (low) | Prospective | HIV-infected and HIV-uninfected patients | 75% | 49 | Adults | 44/5 | 25/24 | microbiologically confirmed TB | 100% PTB | rHBHAms | ELISA | 16–20 | 0.25IU/mL | Whole blood |
| Sali 2018 [22] | NA | Italy (low) | Prospective | Contact individuals and patients | 30% | 64 | Children | 36/28 | 19/45 | microbiologically confirmed TB | 26% PTB | rHBHAms | ELISA | 16–24 | 0.25IU/mL | Whole blood |
| Tang 2020 [23] | 2019.08–2019.12 | China (high) | Prospective | HCWs and patients | 80% | 62 | Adults | 36/26 | 40/22 | microbiologically confirmed TB | 100% PTB | rHBHAms | ELISA | 18 | 22.4pg/mL | Whole blood |
| Dirix 2016 [24] | 2008.02–2010.05 | Uganda (high) | Prospective | HIV-infected and HIV-uninfected patients | NA | 147 | Adults | NA | 62/85 | microbiologically confirmed TB | NA | nHBHA | ELISA | 72 | 75pg/mL | PBMCs |
| Delogu 2016 [25] | 2011.12–2014.04 | Italy (low) | Prospective | HIV-infected patients | 78% | 41 | Adults | 35/6 | 18/23 | microbiologically confirmed TB | NA | rHBHAms | ELISA | 72 | 0.25IU/mL | Whole blood |

Note: In Delogu 2011 [a], the results were from 24-hour antigen stimulation; In Delogu 2011 [b], the results were from 168-hour antigen stimulation.

Abbreviations: NA, not available; HCWs, healthcare workers; BCG, Bacillus Calmette-Guérin; LTBI, latent tuberculosis Infection; TB, tuberculosis; PTB, pulmonary tuberculosis; EPTB, extra-pulmonary tuberculosis; nHBHA, native HBHA; rHBHAms, recombinant HBHA purified from *Mycobacterium smegmatis*; IFA, immunofluorescence assay; ELISA, enzyme linked immunosorbent assay; ELISPOT, Enzyme Linked Immunospot Assay; IU, international unit; SFCs, spots forming cells; PBMCs, Peripheral blood mononuclear cells.

**Table 2. Diagnostic performance of the HBHA-IGRA for discrimination of the LTBI and active TB.**

| Study | Sample size | TP | FP | FN | TN | Sensitivity (95%CI) | Specificity (95% CI) |
|---|---|---|---|---|---|---|---|
| Wyndham-thomas 2014 | 49 | 24 | 6 | 8 | 11 | 0.75 (0.57–0.89) | 0.65 (0.38–0.86) |
| Wen 2017 | 101 | 10 | 17 | 5 | 69 | 0.67 (0.38–0.88) | 0.80 (0.70–0.88) |
| Temmerman 2004 | 101 | 45 | 8 | 10 | 38 | 0.82 (0.69–0.91) | 0.83 (0.69–0.92) |
| Tang 2020 | 62 | 19 | 7 | 3 | 33 | 0.86 (0.65–0.97) | 0.82 (0.67–0.93) |
| Sali 2018 | 64 | 39 | 7 | 6 | 12 | 0.87 (0.73–0.95) | 0.63 (0.38–0.84) |
| Molicotti 2015 | 83 | 43 | 4 | 13 | 23 | 0.77 (0.64–0.87) | 0.85 (0.66–0.96) |
| Molicotti 2011 | 63 | 19 | 7 | 4 | 33 | 0.83 (0.61–0.95) | 0.82 (0.67–0.93) |
| Masungi 2002 | 49 | 15 | 1 | 10 | 23 | 0.60 (0.39–0.79) | 0.96 (0.79–1.00) |
| Hougardy 2007 | 149 | 58 | 48 | 5 | 38 | 0.92 (0.82–0.97) | 0.44 (0.33–0.55) |
| Dirix 2016 | 147 | 21 | 14 | 64 | 48 | 0.25 (0.16–0.35) | 0.77 (0.65–0.87) |
| Delogu 2016 | 41 | 6 | 2 | 17 | 16 | 0.26 (0.10–0.48) | 0.89 (0.65–0.99) |
| Delogu 2011 [a] | 72 | 15 | 13 | 5 | 39 | 0.75 (0.51–0.91) | 0.75 (0.61–0.86) |
| Delogu 2011 [b] | 87 | 13 | 12 | 13 | 49 | 0.50 (0.30–0.70) | 0.80 (0.68–0.89) |
| Chiacchio 2017 | 49 | 13 | 6 | 11 | 19 | 0.54 (0.33–0.74) | 0.76 (0.55–0.91) |
| Combined | | | | | | 0.70 (0.57–0.80) | 0.78 (0.71–0.84) |

Note: In Delogu 2011 [a], the results were from 168-hour antigen stimulation; in Delogu 2011 [b], the results were from 24-hour antigen stimulation.

Abbreviations: TP, true positive; FP, false positive; FN, false negative; TN, true negative; CI, confidence interval.

regression suggested that the inclusion of HIV-infected people is the primary factor leading to the heterogeneity (RDOR = 10.05, 95% CI: 2.62~38.53, $p$ = 0.003). The subgroup analysis (Table 3) showed that the studies which enrolled HIV-infected people revealed much lower sensitivity than the studies unenrolled HIV-infected people ($I^2$ = 87%, $p<0.001$), and the studies which ATB group enrolled the microbiologically and clinically confirmed patients also revealed a higher sensitivity and lower specificity result than the studies which only enrolled the microbiologically confirmed TB patients ($I^2$ = 62%, $p$ = 0.007). Different HBHA antigens, samples for IGRA test, TB burden, and stimulation time did not significantly affect the discrimination accuracy of the HBHA-IGRAs.

### 3.5. Sensitivity analysis

To further examine the impact of individual study on the pooled results, we performed sensitivity analysis (Fig 6). The results of Hougardy et al (2007) [15] and Dirix et al (2016) [24] greatly affect the pooled results (Fig 6C and 6D). After the exclusion of the two studies, the $I^2$ values for heterogeneity were decreased to 76.4% for sensitivity, 13.7% for specificity, 0% for positive DLR, 78.7% for negative DLR, 33.7% for the diagnostic score, and 85.1% for DOR, respectively. Conversely, the sensitivity and specificity had minimal changes, the diagnostic odds ratio (DOR) increased from 8.11 to 9.86. The outcomes of the sequential exclusion of each study (S1 Fig) showed that the DORs did not change significantly in all models, thereby indicating that our results are stable and reliable.

### 4. Discussion

Accurate and early identification of TB infection status is of great significance for reducing the global TB incidence. Because *Mtb* infection causes chronic disease and its clinical symptoms are often atypical, immunodiagnostic tests such as TST and IGRAs (QFT and T-SPOT) are commonly used for screening. The culture, nucleic acid amplification testing (NAAT), and

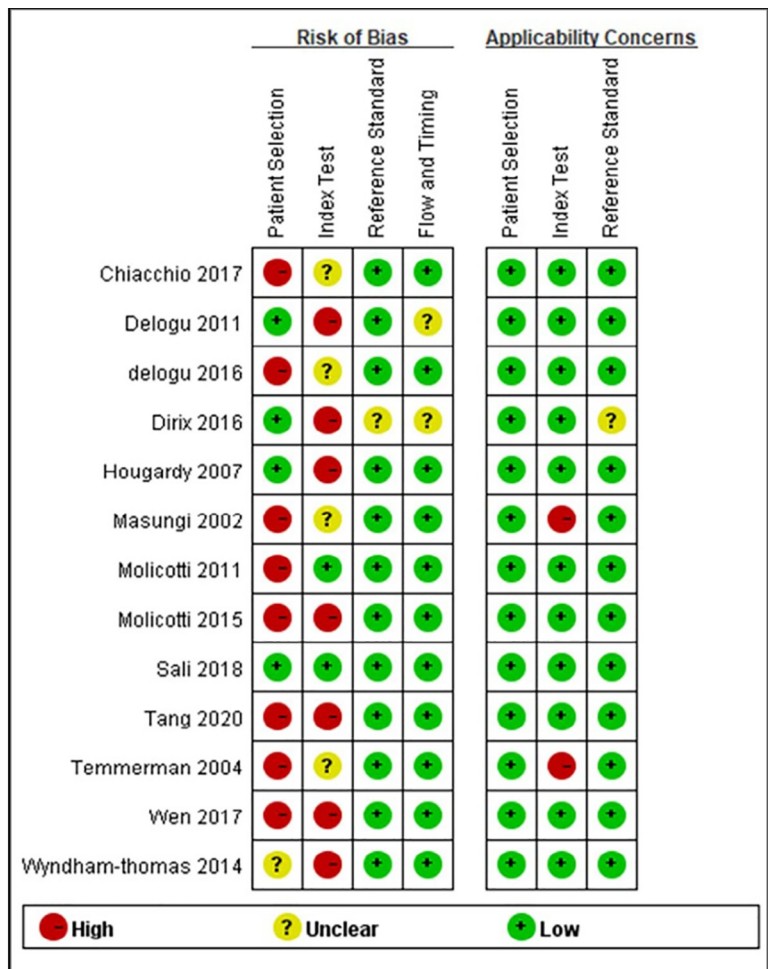

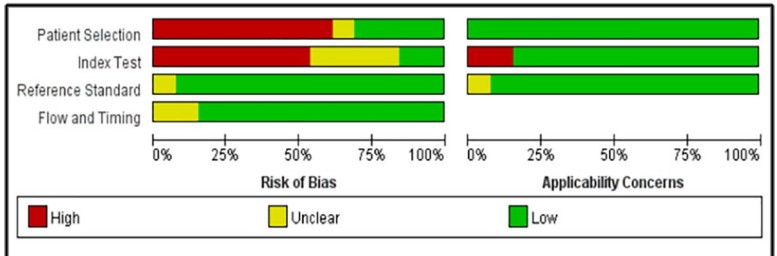

**Fig 2. Summary of articles included regarding the risk of bias and applicability concerns.**

GeneXpert can be used for confirmative diagnosis of TB in clinical practice [26, 27]. The whole process is not only expensive and time-consuming, but also affects timely isolation and treatment, and causes the spread of the *Mtb* infection. Since there are no early and accurate diagnostic tests currently available for detecting active TB and differentiate it from LTBI, immunodiagnostic biomarkers are urgently needed to monitor the progression from LTBI to clinical disease [9, 28, 29]. Studies [9, 10, 11, 30, 31] showed that one of the most promising biomarkers is HBHA. Although the existing studies explored the value of using IGRA with HBHA as a stimulating antigen to differentiate ATB from LTBI, they showed various results and it was difficult to obtain a consensus in deriving an accurate differential diagnosis [15, 16, 20, 22, 32].

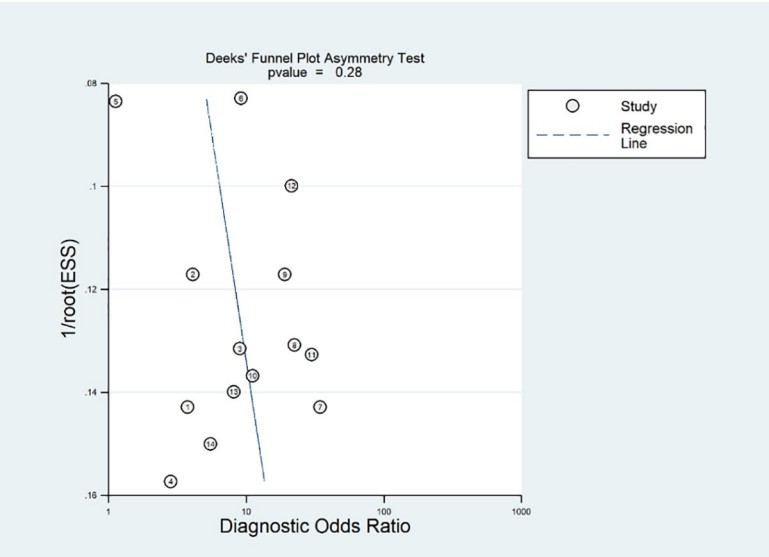

**Fig 3. Deeks' funnel plot asymmetry test.** Non-significant slope indicates that no significant bias was found. ESS; Effective sample size.

This is the first systematic review and meta-analysis on the use of HBHA antigen for the differential diagnosis of ATB and LTBI. No publication bias was detected in any of the studies included in this meta-analysis. This analysis revealed that HBHA-IGRA has acceptable accuracy to differentiate ATB from LTBI in people with a normal T-cell response [sensitivity of 0.78 (95% CI: 0.71–0.85) and specificity of 0.77 (95% CI: 0.70–0.84)], and the Fagan plot also demonstrated satisfactory clinical utility (S2 Fig). However, subgroup analysis showed that the sensitivity of HBHA-induced IFN-γ release in the LTBI subjects was strongly decreased by HIV infection [sensitivity of 0.32 (95% CI: 0.16–0.47) and specificity of 0.79 (95% CI: 0.66–0.92)]. This outcome indicated that low CD4+ T cell number might make it impossible for the HBHA-IGRA to differentiate LTBI from active TB in HIV-infected patients. Interestingly, a current study [33] evaluated the performance of the HBHA-IGRA in HIV-infected individuals living in a low TB incidence country and found that some HIV-infected patients had high responses in contrast to that reported for non-HIV infected subjects; HBHA-IGRA could be more sensitive than both TST and QFT test to identify potentially *Mtb*-infected people. Several studies [13, 14, 18, 34] also demonstrated that both CD4+ and CD8+ T lymphocytes play major roles in IFN-γ synthesis induced by HBHA. Thus, we considered that the reasons for the low sensitivity of the HBHA-IGRA in immunocompromised people are complex and the mechanism of HIV infection affecting HBHA-induced IFN-γ release needs further investigation.

Furthermore, most of the studies did not provide the information on the number of people inoculated with BCG in the LTBI and ATB groups, resulting in the inability to conduct subgroup analysis and deduce the impact of BCG vaccination on the results of this meta-analysis. However, based on the studies [13, 15, 18, 35] investigating the potential impact of a previous BCG vaccination on the HBHA-IGRA results by testing LTBI subjects and healthy controls who can provide accurate information about their BCG vaccination status and combined with the practice of HBHA-IGRA in high TB burden countries such as China [20, 23], it may be concluded that a BCG vaccination before the HBHA-IGRA has no influence on the results.

Regarding the implementation of HBHA-IGRA, the molecular form and the concentration of HBHA are crucial for the sensitivity of the detection of LTBI subjects based on their

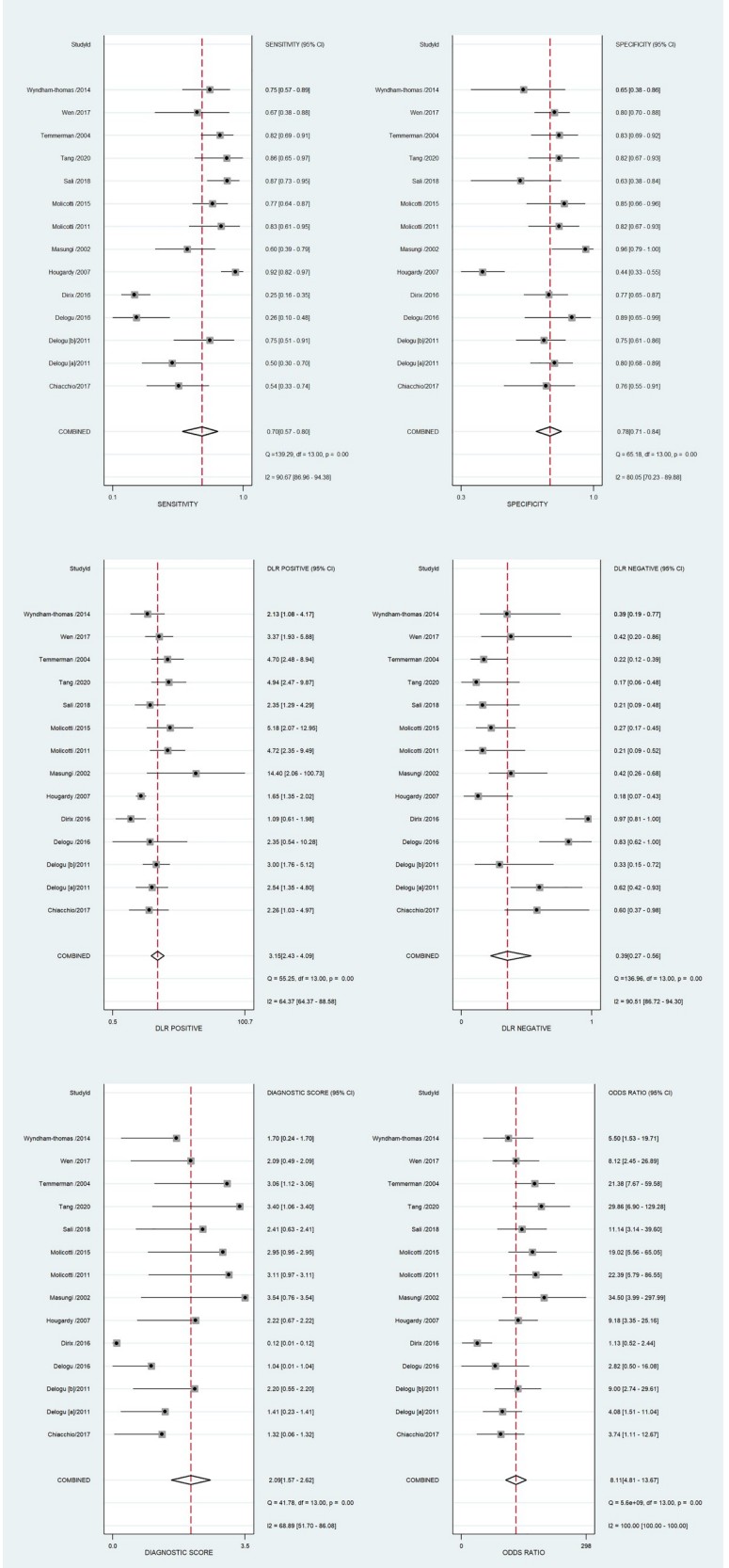

**Fig 4.** Forest plots showing the estimates of (a) sensitivity and specificity, (b) positive likelihood ratio and negative likelihood ratio, and (c) diagnostic score and diagnostic odds ratio (DOR) of the HBHA-IGRA for discrimination of the LTBI and active TB. HBHA-IGRA; mycobacterial heparin-binding hemagglutinin-induced interferon-gamma release assay.

positivity in the HBHA-IGRA test [35]. On the one hand, the IFN-γ secretion induced by native HBHA (nHBHA) has a good relationship with the recombinant HBHA purified from *Mycobacterium smegmatis* (rHBHA-Ms) [14]. On the other hand, T cells from the LTBI subjects who showed no or low IFN-γ response to rHBHA-Ms frequently responded to nHBHA [14, 35]. Therefore, even if no statistically significant difference was detected while using nHBHA or rHBHA-Ms as the stimulating antigen in IGRA format in differentiating the ATB from the LTBI ($p = 0.47$), it is crucial to use the optimal form and the concentration of the antigen. Moreover, the current results also revealed no statistical difference was observed in HBHA-IGRA with respect to the differential diagnosis of ATB and LTBI between the sample sources [peripheral blood mononuclear cells (PBMCs) or whole blood] and between the durations of the IGRA (within or more than 24 hours) ($p = 0.61$ and $p = 0.52$, respectively).

The cut-off values used by all the teams in these studies were different but mainly based on the threshold provided by the two teams from Belgium (100pg/mL) and Italy (0.25 IU/mL). This might be caused by the different monoclonal antibodies, the different reference standards and etc. used in the ELISA process. Nevertheless, upon analysis, no bias caused by the threshold was found in this study. In order to establish a commercial HBHA-IGRA kit, it is important to set an appropriate diagnostic threshold (Cut-off value) and effective reference ranges of people in different TB infection status in the future.

Our meta-analysis has some limitations. First, almost all the included studies were case-control studies. A divergence was noted in the definition of the ATB groups between microbiologically confirmed TB and clinically confirmed TB. The inclusion of the LTBI groups had different selection criteria (TST or IGRA): the TST results are more often positive than the IGRAs, the LTBI groups in different studies exhibited varied risk stratifications [11, 30, 36, 37]. Moreover, some studies specified neither the active TB type (pulmonary or extra-pulmonary TB) of the ATB subjects nor the treatment of the LTBI subjects. Hence, a potential heterogeneity could be present in the target population, resulting in the poor quality of "patient selection". The variability of the cut-off between different studies might also be due to different demographic characteristics of the populations included in the studies. Next, we only found heterogeneity between normal people and HIV-infected people, but there is still heterogeneity in sensitivity, negative DLR, and DOR among the studies after the exclusion. Therefore, we suspect that the influencing factors included for the subgroup analysis were insufficient. Finally, although a large number of studies were screened for this review, only 13 studies were included in the final analysis. The main reason that led to several exclusions, was the lack of detailed test results using HBHA-IGRA to detect both ATB and LTBI groups. Further, six of 13 included studies were performed in Italy and four of these studies came from Belgium. This may also be a reason that why the conclusion drawn by this meta-analysis is biased.

In conclusion, the results of this meta-analysis suggest that the HBHA-IGRA can be a good diagnostic tool for the discrimination of the latent and active TB, and combination of the results of the HBHA-IGRA with those from other IGRAs (ESAT-6 and CFP-10-based) may allow optimal stratification of *Mtb* infected patients in different groups with variable risks of reactivation of the infection. Currently, the HBHA-IGRA is the only promising IGRA test discriminating between active TB and LTBI. However, due to the lack of large and high-quality studies in high TB burden countries and immune dysfunction people, the application conditions of HBHA-IGRA need to be clarified further. In order to commercialize the HBHA-

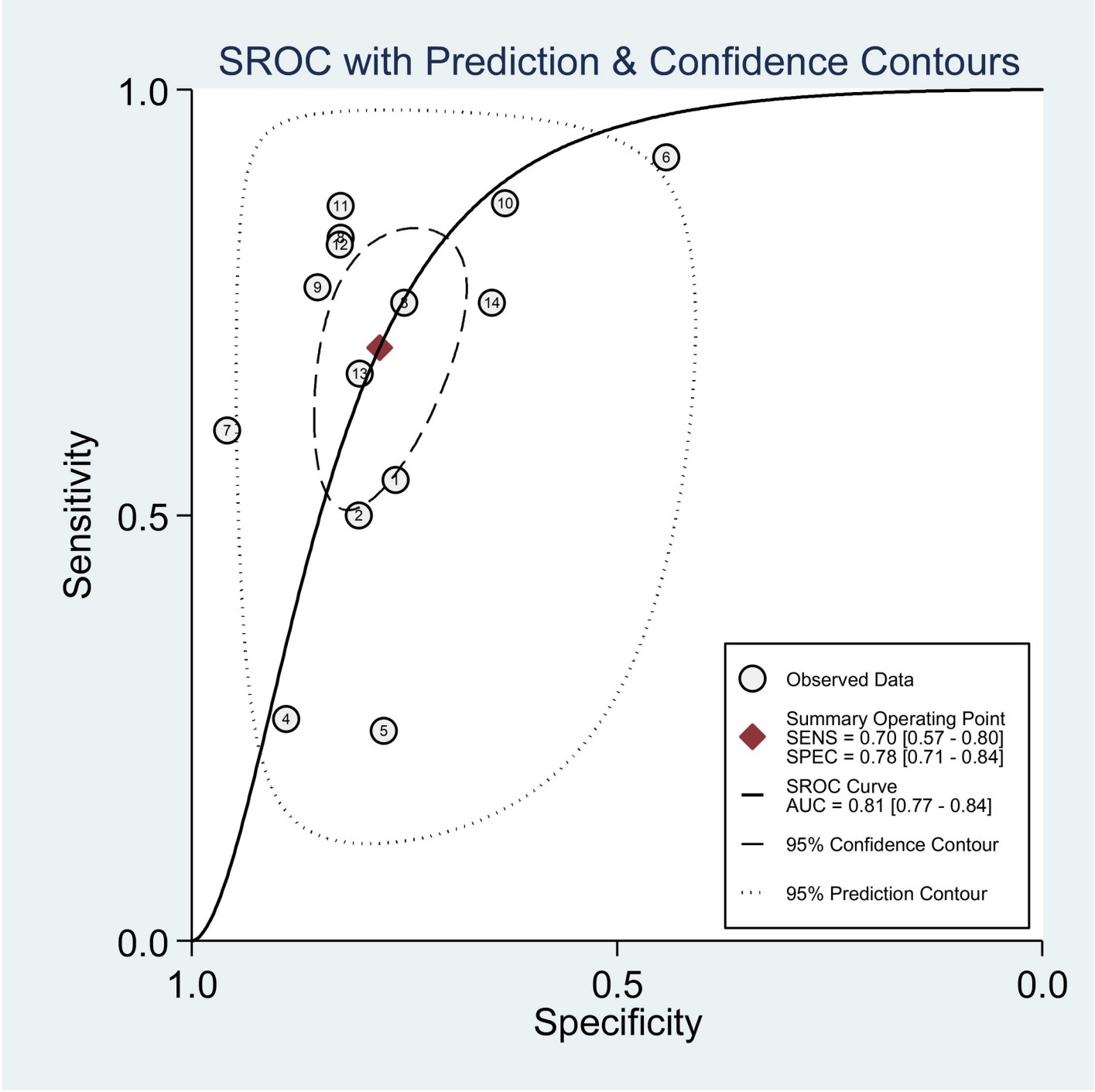

**Fig 5. Summary receiver operating characteristic (ROC) curves of the HBHA-IGRA for discrimination of the LTBI and active TB.** HBHA-IGRA; mycobacterial heparin-binding hemagglutinin-induced interferon-gamma release assay.

based IGRA to efficiently distinguish LTBI from ATB, it is urgent to raise the interest of commercial companies in this test to provide kits with well-defined technical conditions and cut-off values.

**Table 3. Subgroup analysis of the HBHA-IGRA for discrimination of the LTBI and active TB.**

| Covariate | Subgroup | N | Meta-analytic summary estimate | | | |
|---|---|---|---|---|---|---|
| | | | Sensitivity (95%CI) | Specificity (95%CI) | $I^2$ (%) | p value |
| Population | HIV-infected enrolled | 3 | 0.32 (0.16–0.47) | 0.79 (0.66–0.92) | 87 | 0.00* |
| | HIV-infected unenrolled | 11 | 0.78 (0.71–0.85) | 0.77 (0.70–0.84) | | |
| TB burden | High | 3 | 0.58 (0.29–0.87) | 0.80 (0.68–0.92) | 0 | 0.64 |
| | Low | 11 | 0.72 (0.60–0.85) | 0.77 (0.70–0.85) | | |
| Active TB definition | microbiologically confirmed TB | 11 | 0.66 (0.52–0.80) | 0.81 (0.76–0.87) | 62 | 0.07 |
| | clinically confirmed TB | 3 | 0.81 (0.63–0.99) | 0.64 (0.51–0.77) | | |
| Antigen type | nHBHA | 5 | 0.70 (0.51–0.90) | 0.73 (0.62–0.84) | 0 | 0.47 |
| | rHBHAms | 9 | 0.70 (0.55–0.85) | 0.80 (0.73–0.87) | | |
| Sample type | PBMCs | 6 | 0.70 (0.51–0.88) | 0.75 (0.65–0.84) | 0 | 0.61 |
| | whole blood | 8 | 0.70 (0.54–0.86) | 0.80 (0.73–0.88) | | |
| Stimulation time | >24h | 6 | 0.63 (0.44–0.82) | 0.77 (0.68–0.87) | 0 | 0.52 |
| | ≤24h | 8 | 0.74 (0.60–0.88) | 0.78 (0.70–0.86) | | |

Abbreviations: TB, tuberculosis; nHBHA, native HBHA; rHBHAms, recombinant HBHA purified from *Mycobacterium smegmatis*; PBMCs, Peripheral blood mononuclear cells.

*, $p < 0.05$.

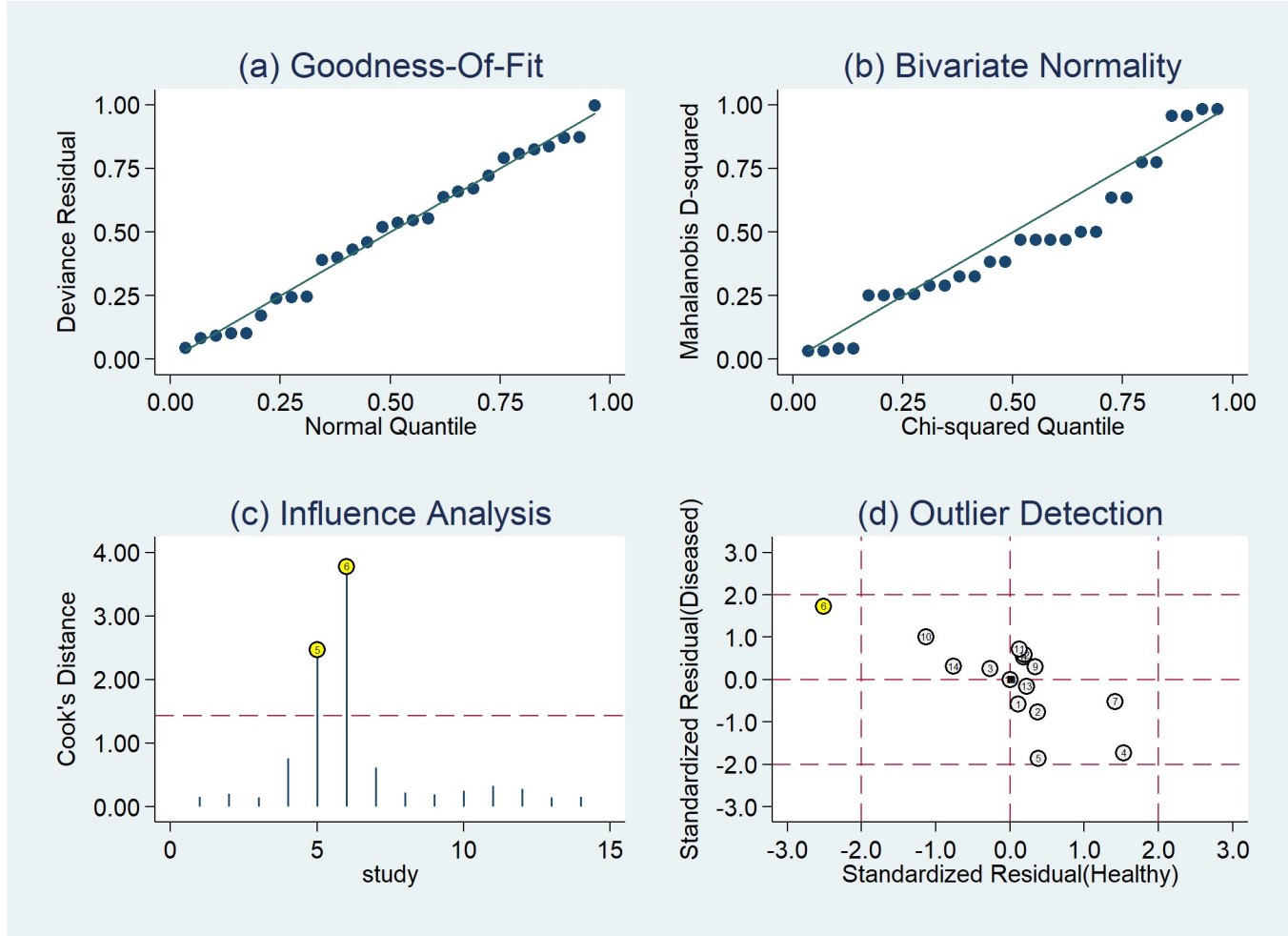

**Fig 6.** The results of sensitivity analysis (a: Goodness of fit. b: Bivariate normality. c: Influence analysis. d: Outlier detection).

## Supporting information

**S1 Checklist.**
(DOC)

**S1 Fig. Meta-analysis estimates of diagnostic odds ratio (DOR) of the HBHA-IGRA for discrimination of the LTBI and active TB, through deleting each study one by one.**
(TIF)

**S2 Fig. Fagan plot to evaluate the clinical utility of the HBHA-IGRA for discrimination of the LTBI and active TB.** Pre-test probability = 50%.
(TIF)

## Author Contributions

**Conceptualization:** Jinhua Tang, Yueyun Ma.

**Data curation:** Jinhua Tang, Yuan Huang.

**Formal analysis:** Jinhua Tang, Yuan Huang.

**Funding acquisition:** Yueyun Ma.

**Investigation:** Jinhua Tang, Zheng Cai.

**Methodology:** Jinhua Tang, Yuan Huang, Zheng Cai.

**Project administration:** Jinhua Tang, Yueyun Ma.

**Resources:** Jinhua Tang.

**Software:** Jinhua Tang.

**Supervision:** Yueyun Ma.

**Validation:** Jinhua Tang, Yueyun Ma.

**Visualization:** Jinhua Tang, Yueyun Ma.

**Writing – original draft:** Jinhua Tang.

**Writing – review & editing:** Yuan Huang, Zheng Cai, Yueyun Ma.

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
