## [Decision Letter · Decision Letter 0]

28 Apr 2021

PONE-D-21-06709

Mycobacterial heparin-binding hemagglutinin (HBHA)-induced interferon-γ release assay (IGRA) for discrimination of latent and active tuberculosis: a systematic review and meta-analysis

PLOS ONE

Dear Dr. Ma,

Thank you for submitting your manuscript to PLOS ONE. After careful consideration, we feel that it has merit but does not fully meet PLOS ONE’s publication criteria as it currently stands. Therefore, we invite you to submit a revised version of the manuscript that addresses the points raised during the review process.

We look forward to receiving your revised manuscript.

Kind regards,

Olivier Neyrolles

Academic Editor

PLOS ONE

Reviewers' comments:

Reviewer's Responses to Questions

**Comments to the Author**

1. Is the manuscript technically sound, and do the data support the conclusions?

Reviewer #1: Partly

Reviewer #2: Yes

Reviewer #3: Yes

2. Has the statistical analysis been performed appropriately and rigorously? 

Reviewer #1: No

Reviewer #2: Yes

Reviewer #3: I Don't Know

3. Have the authors made all data underlying the findings in their manuscript fully available?

Reviewer #1: Yes

Reviewer #2: Yes

Reviewer #3: Yes

4. Is the manuscript presented in an intelligible fashion and written in standard English?

Reviewer #1: Yes

Reviewer #2: Yes

Reviewer #3: Yes

5. Review Comments to the Author

Reviewer #1: This manuscript reports summarized results for HBHA-induced interferon-γ release assay (IGRA) for discrimination of latent and active tuberculosis using a systematic review and meta-analysis. In overall the statistical analysis follows the routine procedures for diagnostic meta-analysis. I have below comments.

The heterogeneity is high, it is not useful to simply pool the results. It will be informative to look widely and deeply about the resources for the variation and provide corresponding discussions. E.g. Data from Dirix 2016 and Delogu 2016 are quite different from other cited studies. Can you investigate the potential reasons for the observed difference?

For each measurement, it will be informative to add sensitivity analysis to find which study would mostly affect the pooled results and provide discussion.

Besides diagnostic odds ratio, please also add analysis to examine publication bias for other measures.

Reviewer #2: The manuscript is interesting and well written.

page 9, row 56: …”recommends that an interferon-γ (IFN-γ) release assay “please change as …recommends that an interferon (IFN)-γ release assay

TABLE 2: please, complete the title as Diagnostic performance of HBHA-IGRA or…LTBI discrimination? Moreover in the footnote, please explain the abbreviations: TP FP FN TN

Reviewer #3: The clinical use of the HBHA-IGRA to differentiate active TB from LTBI has been reported in several different studies. However the sample size in each study is most often limited and the HBHA-IGRA used in different publications present several technical differences, most importantly the nature of the HBHA antigen (native versus recombinant).

In this manuscript, the authors performed a systematic review of the published results to summarize the current state of this research, and they performed a meta-analysis of the published results to determine the efficacy of the HBHA-IGRA to differentiate active from latent TB for its clinical utilization.

Even if the subject is of great interest, several points need to be addressed before publication of this meta-analysis as detailed below.

Introduction

Lines 66-67 the review by Mascart F, Locht C published in 2015 in Exp Rev Vacc should be added as a ref for the discriminatory potential of HBHA between active and latent TB.

Material and methods

Study selection criteria

1. The authors indicate that the LTBI subjects were selected on the basis of a TST> 10 mm or a positive IGRA in absence of symptoms or signs of active TB.

As numerous studies reported a poor correlation between the TST results and those from IGRA, this means that the patients selection is quite different if it is based on TST results or on IGRA results. TST results are most often positive than IGRA so that different authors identify at least two different groups within TST+ LTBI, those with a positive IGRA and those with a negative IGRA (Corbiere C et al Plos One 2012; Mascart F et al Exp Rev Vacc 2015; Serrano CJ et al Clin Immunol 2015). These differences should be mentioned and discussed.

Moreover, most studies did not clearly indicated if both untreated and treated LTBI subjects were included and this point should also be mentioned as a limitation and cause of heterogeneity of the data.

2. The ATB groups is indicated to be defined as microbiologically confirmed TB patients, a definition that is quite restrictive and probably not applied by some authors. In clinical situations, diagnosis of ATB is accepted either in case of microbiological confirmation or in case of high clinical suspicion and objectivated clinical response to treatment. This heterogeneity should be mentioned.

Another heterogeneity among patients with ATB is the presence in most studies of both pulmonary and extra-pulmonary ATB and this heterogeneity should be mentioned when available for instance in Table 1, as results of the HBHA-IGRA are not necessarily similar in both forms of active TB.

Results

3.3 lines 176-177: how was the diagnostic accuracy of the HBHA-IGRA for discrimination between active and latent TB calculated when it is not provided in the original manuscript?

Same question applied for the likehood ratio, Odds ratio and diagnostic scores.

Minor comment: the same order should applied for the different studies in Table 2 and Fig 4

Discussion

Lines 212-213 – studies showed that one of the most promising biomarkers to differentiate active from latent TB is HBHA. Two ref should be added here: Mascart F , Locht C Exp Rev Vacc 2015 and Smits K et al Clin Cell Immunol 2015

Line 222: low sensitivity in immunocompromised patients. Is it not mostly / only HIV infected people? If yes, this should be mentioned.

Line 225: the authors suggest that the release of HBHA-induced IFN-g strongly depends on CD4+ T lymphocytes as its sensitivity is low in immunocompromised patients. In fact, several papers clearly demonstrated that both CD4+ and CD8+ T lymphocytes play a major role in the IFN-g synthesis induced by HBHA and this should be mentioned (ref 12-13-17). This means that the link with a lower sensitivity in immunocompromised people is not so clear for the moment and should be further investigated.

Line 229-230 – possible interference of BCG vaccination. If most studies did not provide detailed information about the BCG status of the patients, this does not mean that we cannot conclude about the possible interference of BCG/ or not on the results of the HBHA-IGRA. It is indeed often difficult to know the BCG status from patients with ATB as most often these patients do not remain themselves if they were vaccinated or not. However, several studies investigated the possible impact of a previous BCG vaccination on the HBHA-IGRA results by testing subjects with LTBI and most importantly healthy controls who can provide more consistent information about their BCG vaccination status. By doing so, it may be clearly concluded that a BCG vaccination administrated at least 10 years before the HBHA-IGRA has no influence on the results (see ref 12- 14-17-31). This point is important to mention in the discussion.

Lines 232-234: “native HBHA (nHBHA) showed better IFN-γ release than the

recombinant HBHA purified from Mycobacterium smegmatis (rHBHA-Ms) upon

peptide stimulation, »

“upon peptide stimulation” should be deleted as this is not correct. It is upon stimulation with either nHBHA or rHBHA-Ms.

Importantly, these differences between the two HBHA preparations were shown to be important for the sensitivity of detection of LTBI subjects by their positivity in the HBHA-IGRA. This paper (ref 31) does not concern differential diagnosis between active and latent TB. It should be indicated that even if the molecular form of HBHA was reported to be important for an accurate detection of LTBI, nHBHA being more sensitive that rHBHA-Ms, this is probably not the case for the differential diagnosis between LTBI and aTB as suggested by the meta-analysis reported here.

Line 243: “the best threshold should be selected”. This is impossible to do as different ELISA tests are used with different monoclonal antibodies, different standards etc….(same comment applies for line 252)

Conclusions

The main conclusion from this meta-analysis is that the HBHA-IGRA is a promising tool to differentiate active from latent TB. However, several publications indicated that an even better discrimination may be obtained by combining the results of the HBHA-IGRA to those of an ESAT-6-IGRA (or QFT ) (see for instance ref 14) and this point should be added to the discussion.

Line 260: we may not conclude that the HBHA-IGRA is a good diagnostic tool in high TB burden countries as only a low number of studies were performed in these countries.

Line 261: “the application of the HBHA-IGRA is restricted by the immune status “ Here again, there are too few studies to draw such a conclusion. Studies on the performance of the HBHA-IGRA to detect LTBI (and not for differential diagnosis between ATB and LTBI) among patients under hemodialysis have shown a high diagnostic accuracy of the HBHA-IGRA test which was more sensitive than the TST and the QFT (Dessein R et al PLoS One 2013).

Lines 262-264. If the HBHA-IGRA tests still has to be standardized for a commercial application, it should however be acknowledged that it is the only IGRA test providing actually such a good discrimination between ATB and LTBI. It is therefore now quite urgent that commercial companies became interested to introduce this test in the commercially available portofolio of standardized IGRA.

6. PLOS authors have the option to publish the peer review history of their article (what does this mean?). If published, this will include your full peer review and any attached files.

Reviewer #1: No

Reviewer #2: No

Reviewer #3: **Yes: **Françoise Mascart

---

## [Author Response · Author response to Decision Letter 0]

24 May 2021

Dear Dr. Olivier Neyrolles and Reviewers:

Thank you for your letter and for the reviewers’ comments concerning our manuscript entitled “Mycobacterial heparin-binding hemagglutinin (HBHA)-induced interferon-γ release assay (IGRA) for discrimination of latent and active tuberculosis: a systematic review and meta-analysis” (Manuscript Number: PONE-D-21-06709). These comments are all valuable and very helpful for improving the manuscript. We have studied comments carefully and have made corrections which we hope meet with approval. The main revised portion are marked in the Revised Manuscript with Track Changes. The responds to the reviewer’s comments are as follows:

Reviewer #1: This manuscript reports summarized results for HBHA-induced interferon-γ release assay (IGRA) for discrimination of latent and active tuberculosis using a systematic review and meta-analysis. In overall the statistical analysis follows the routine procedures for diagnostic meta-analysis. I have below comments.

Comments for the author

1. The heterogeneity is high; it is not useful to simply pool the results. It will be informative to look widely and deeply about the resources for the variation and provide corresponding discussions. E.g. Data from Dirix 2016 and Delogu 2016 are quite different from other cited studies. Can you investigate the potential reasons for the observed difference?

Response: Thanks for your comments. Yes, the heterogeneity of the pooled results is high. In order to explore the resources for the variation, we conducted meta regression and subgroup analysis based on the characteristics of the included studies. Through subgroup analysis, we found that the inclusion of HIV-infected people is the major factor leading to the heterogeneity. Dirix 2016 and Delogu 2016 are both clinical studies involving subjects infected with HIV, so data from Dirix 2016 and Delogu 2016 are quite different from other enrolled studies. According to your suggestion, we conducted a thorough analysis and discussion on the potential reasons for the difference, related revisions are marked in the discussion.

2. For each measurement, it will be informative to add sensitivity analysis to find which study would mostly affect the pooled results and provide discussion.

Response: According to your suggestion, we added the “Sensitivity analysis” to the results to verify the robustness of the findings (Fig. 6). In our results, Hougardy 2007 would mostly affect the pooled results. After the exclusion of this study, the I2 for heterogeneity of specificity decreased from 80.05% (p=0.00) to 20.06% (p=0.24). Simultaneously, Dirix 2016 would mostly affect the pooled diagnostic efficacy. After the exclusion of this study, the I2 for heterogeneity of diagnostic score decreased from 68.9% (p=0.00) to 28.7% (p=0.16), the diagnostic odds ratio (DOR) increased from 8.11 to 9.94. However, through omitting each study one by one (Fig. S1), the outcomes indicated that the DORs did not change significantly in all models, showing that our results are stable and reliable. The related revisions are marked in the paper.

3. Besides diagnostic odds ratio, please also add analysis to examine publication bias for other measures.

Response: Thanks for your suggestion. However, separate funnel plots for sensitivity and specificity (after logit transformation) are unlikely to be helpful for detecting sample size effects, because sensitivities and specificities will vary due to both variability of threshold between the studies and random variability. Simultaneous interpretation of two related funnel plots and two tests for funnel plot asymmetry also presents difficulties. At present, formal testing for publication bias may be conducted by a regression of lnDOR (Diagnostic Odds Ratio) against 1/ESS1/2, weighting by ESS (Deeks, 2005), with P <0.05 for the slope coefficient indicating significant asymmetry. For the meta-analysis of the accuracy of diagnostic tests, the examination method of publication bias is limited to Deeks’ test. Other methods such as Egger, Begg, Harbord and Peters tests commonly used in intervention studies are not suitable here due to the high false positive rate.

Reviewer #2: The manuscript is interesting and well written.

Comments for the author

1. Page 9, row 56: “recommends that an interferon-γ (IFN-γ) release assay” please change as “recommends that an interferon (IFN)-γ release assay”.

Response: Thank you for your valuable comments. We changed the words. The related revised portion is marked in red in the paper.

2. TABLE 2: please, complete the title as Diagnostic performance of HBHA-IGRA or…LTBI discrimination? Moreover, in the footnote, please explain the abbreviations: TP FP FN TN.

Response: The imperfect titles of TABLE 2 and TABLE 3 are completed in the paper, and the related abbreviations in the footnote of TABLE 2 is added.

Reviewer #3: The clinical use of the HBHA-IGRA to differentiate active TB from LTBI has been reported in several different studies. However, the sample size in each study is most often limited and the HBHA-IGRA used in different publications present several technical differences, most importantly the nature of the HBHA antigen (native versus recombinant). In this manuscript, the authors performed a systematic review of the published results to summarize the current state of this research, and they performed a meta-analysis of the published results to determine the efficacy of the HBHA-IGRA to differentiate active from latent TB for its clinical utilization. Even if the subject is of great interest, several points need to be addressed before publication of this meta-analysis as detailed below.

Comments for the author

1. Introduction:

Lines 66-67 the review by Mascart F, Locht C published in 2015 in Exp Rev Vacc should be added as a ref for the discriminatory potential of HBHA between active and latent TB.

Response: Thank you very much for your detailed and precious comments. We read this review carefully and benefited a lot. The reference has been added in this part.

2. Material and methods:

Study selection criteria

2.1. The authors indicate that the LTBI subjects were selected on the basis of a TST> 10 mm or a positive IGRA in absence of symptoms or signs of active TB. As numerous studies reported a poor correlation between the TST results and those from IGRA, this means that the patient’s selection is quite different if it is based on TST results or on IGRA results. TST results are most often positive than IGRA so that different authors identify at least two different groups within TST+ LTBI, those with a positive IGRA and those with a negative IGRA (Corbiere V et al Plos One 2012; Mascart F et al Exp Rev Vacc 2015; Serrano CJ et al Clin Immunol 2015). These differences should be mentioned and discussed. Moreover, most studies did not clearly indicate, if both untreated and treated LTBI subjects were included and this point should also be mentioned as a limitation and cause of heterogeneity of the data.

Response: We very much agree with your suggestion that the LTBI subjects should be further stratified based on the results of TST and IGRA results. We reviewed all published studies of the clinical use of the HBHA-IGRA to differentiate active TB from LTBI, most studies used “a positive IGRA result” as the selection criteria for the LTBI subjects, and a small number of early studies used “healthy people with a TST≥10 mm” as the criterion for the LTBI subjects. However, it is not clear from enrolled articles whether the LTBI subjects had both received the TST and IGRA tests and whether they received treatment. At the same time, although the patient’s selection is quite different whether it is based on TST results or on IGRA results, the incidence of active TB is still substantial in numerous at-risk populations after a positive TST or IGRA result (Campbell JR et al BMJ 2020). Therefore, in order to analyze as comprehensively as possible and avoid excessive exclusion, we defined the LTBI subjects as individuals with a TST≥10 mm or a positive IGRA in absence of symptoms or signs of active TB but were at risk for the active TB. Certainly, the difference in the enrollment of LTBI subjects due to the different IGRA screening tests (TST and IGRA) and the difference between untreated and treated LTBI subjects must also be added in the limitation and the resources of heterogeneity of the data. The related revised portion is marked in the paper.

2.2. The ATB groups is indicated to be defined as microbiologically confirmed TB patients, a definition that is quite restrictive and probably not applied by some authors. In clinical situations, diagnosis of ATB is accepted either in case of microbiological confirmation or in case of high clinical suspicion and objectivated clinical response to treatment. This heterogeneity should be mentioned. Another heterogeneity among patients with ATB is the presence in most studies of both pulmonary and extra-pulmonary ATB and this heterogeneity should be mentioned when available for instance in Table 1, as results of the HBHA-IGRA are not necessarily similar in both forms of active TB.

Response: Based on your suggestion, we added the “Active TB definition” part and the “Active TB type” part in Table 1. Through subgroup analysis (added in Table 3), we found the studies which defined ATB groups as microbiologically confirmed TB patients tend to have lower sensitivity and higher specificity than the studies which active TB definition was based either on microbiological proof or on high clinical suspicion with favorable response to anti-TB treatment. The heterogeneity is high (I2=62%). However, although the ATB subjects in most studies included both pulmonary and extra-pulmonary ATB patients, only four of the 13 enrolled studies provided relevant data. Therefore, as a possible source of heterogeneity, it can only be elaborated in the discussion. The related revised portion is marked in the results and the discussion.

3. Results:

lines 176-177: how was the diagnostic accuracy of the HBHA-IGRA for discrimination between active and latent TB calculated when it is not provided in the original manuscript? Same question applied for the likehood ratio, Odds ratio and diagnostic scores.

Minor comment: the same order should be applied for the different studies in Table 2 and Fig 4

Response: We first searched all original research papers that used the HBHA-IGRA to differentiate active TB from LTBI through a specific search strategy in public databases. As there are a few studies in this field at present, 13 of the 30 related literature were selected for further meta-analysis. In order to avoid inappropriate inclusion and exclusion, we conducted the inclusion criteria for the active TB and LTBI subjects, and selected subjects from each article to be enrolled in our study based on these criteria (not all the active TB and LTBI subjects in each article were included in this meta-analysis). According to the summary charts of the IFN-γ levels induced by HBHA in the ATB and LTBI subjects and the cut-off values of HBHA-IGRA provided in each article, we would able to obtain the number of true positives, false positives, false negatives, and true negatives, and then used this data to construct a diagnostic 2x2 table. Using the model estimated coefficients and variance-covariance matrices, the Stata v.14.0 software could calculate the pooled sensitivity and specificity, summary likelihood and odds ratios, diagnostic scores and Summary Receiver Operating Characteristic Curves (SROC), and the global and relevant test performance metric-specific heterogeneity statistics are also provided. In addition, the same order has been used for the different studies in Table 2 and Fig 4, the related revised portion is marked in the paper. 

4. Discussion:

4.1. Lines 212-213: studies showed that one of the most promising biomarkers to differentiate active from latent TB is HBHA. Two refs should be added here: Mascart F, Locht C Exp Rev Vacc 2015 and Smits K et al Clin Cell Immunol 2015

Response: “Integrating knowledge of Mycobacterium tuberculosis pathogenesis for the design of better vaccines.” (Mascart F, Locht C Exp Rev Vacc 2015) has been added here as a ref. However, we searched the public databases and did not find the other ref (Smits K et al Clin Cell Immunol 2015). We would appreciate it if you could provide more specific information.

4.2. Line 222: low sensitivity in immunocompromised patients. Is it not mostly / only HIV infected people? If yes, this should be mentioned.

Response: Yes. The paper only conducted the subgroup analysis on HIV-infected and uninfected people, so “immunocompromised patients” here is not appropriate. The portion has been revised and marked in the paper.

4.3. Line 225: the authors suggest that the release of HBHA-induced IFN-g strongly depends on CD4+ T lymphocytes as its sensitivity is low in immunocompromised patients. In fact, several papers clearly demonstrated that both CD4+ and CD8+ T lymphocytes play a major role in the IFN-g synthesis induced by HBHA and this should be mentioned (ref 12-13-17). This means that the link with a lower sensitivity in immunocompromised people is not so clear for the moment and should be further investigated.

Response: After HIV infects the human body, the main target cells are CD4+ T lymphocytes. Simultaneously, the sensitivity of the HBHA-IGRA of the HIV-infected enrolled studies is generally low. Our discussion here was to suppose that whether the decline in IFN-γ release induced by HBHA is mainly caused by HIV damage to CD4+ T lymphocytes. It was not intended to show that the release of HBHA-induced IFN-γ only depends on CD4+ lymphocytes. Because the mechanism by HBHA-induced high levels of IFN-γ in LTBI people may be very complicated and remains unclear. Thank you for pointing out our problem. Based on your suggestions, we modify the discussion here, and emphasize the important role of CD4+ and CD8+ T lymphocytes in the IFN-γ synthesis induced by HBHA.

4.4. Line 229-230: possible interference of BCG vaccination. If most studies did not provide detailed information about the BCG status of the patients, this does not mean that we cannot conclude about the possible interference of BCG/ or not on the results of the HBHA-IGRA. It is indeed often difficult to know the BCG status from patients with ATB as most often these patients do not remain themselves if they were vaccinated or not. However, several studies investigated the possible impact of a previous BCG vaccination on the HBHA-IGRA results by testing subjects with LTBI and most importantly healthy controls who can provide more consistent information about their BCG vaccination status. By doing so, it may be clearly concluded that a BCG vaccination administrated at least 10 years before the HBHA-IGRA has no influence on the results (see ref 12- 14-17-31). This point is important to mention in the discussion.

Response: We agree with your opinion. The inability to conduct subgroup analysis does not mean that the conclusion cannot be drawn from the previous studies. The studies you provided and several studies in China where BCG is commonly vaccinated all showed that a previous BCG vaccination had no influence on the efficacy of the HBHA-IGRA in differentiating active TB from LTBI. This point has been added in the discussion.

4.5. Lines 232-234: “native HBHA (nHBHA) showed better IFN-γ release than the recombinant HBHA purified from Mycobacterium smegmatis (rHBHA-Ms) upon peptide stimulation”-“upon peptide stimulation” should be deleted as this is not correct. It is upon stimulation with either nHBHA or rHBHA-Ms.

Importantly, these differences between the two HBHA preparations were shown to be important for the sensitivity of detection of LTBI subjects by their positivity in the HBHA-IGRA. This paper (ref 31) does not concern differential diagnosis between active and latent TB. It should be indicated that even if the molecular form of HBHA was reported to be important for an accurate detection of LTBI, nHBHA being more sensitive that rHBHA-Ms, this is probably not the case for the differential diagnosis between LTBI and aTB as suggested by the meta-analysis reported here.

Response: As you suggested, it is stimulated by purified nHBHA or rHBHA-Ms antigen protein, not the peptides, “upon peptide stimulation” has been deleted. The expression “although a recent study (ref 31) found that the native HBHA (nHBHA) showed better IFN-γ release than the recombinant HBHA purified from Mycobacterium smegmatis (rHBHA-Ms)” in our paper is inaccurate. Compared with rHBHA-Ms, nHBHA is more easily recognized by T cells from latently-infected humans (about 100-fold better), so it has better sensitivity in detection of LTBI subjects, as shown in Corbière V J Immunol 2020 (ref 31). Therefore, even if there is no statistically significant difference of using nHBHA or rHBHA-Ms as stimulating antigen in IGRA format in differentiating the ATB from the LTBI (p=0.47), it is crucial to use the optimal form and concentration of the antigen. The related portion revised according to your suggestion is indeed more in line with the real situation.

4.6. Line 243: “the best threshold should be selected”. This is impossible to do as different ELISA tests are used with different monoclonal antibodies, different standards etc. (same comment applies for line 252)

Response: For studies with small sample size of different teams in different countries, it is indeed impossible to select the best threshold in the meta-analysis. However, for large multi-centre studies aimed at producing mature commercial HBHA-IGRA kits, it is very important to establish effective reference ranges in people with different TB infection status and select the best diagnostic threshold (Cut-off value) based on the clinical test results. The expression in our paper may have caused confusion, and it has been revised.

5. Conclusions:

5.1. The main conclusion from this meta-analysis is that the HBHA-IGRA is a promising tool to differentiate active from latent TB. However, several publications indicated that an even better discrimination may be obtained by combining the results of the HBHA-IGRA to those of an ESAT-6-IGRA (or QFT) (see for instance ref 14) and this point should be added to the discussion.

Response: We agree with your suggestion. As shown in our previous study (Tang JH et al Tuberculosis 2020) and the recent publications (Hougardy JM et al Plos One 2007, Sali M et al J Infection 2018 and Chedid C et al Front Immunol 2020), the combination of the results of the HBHA-IGRA and the commercial IGRAs based on ESAT-6 or CFP-10 (such as QFT, etc.) may indeed have a better discrimination. This point has been added to the discussion in the revised paper.

5.2. Line 260: we may not conclude that the HBHA-IGRA is a good diagnostic tool in high TB burden countries as only a low number of studies were performed in these countries.

Response: Three of the 13 included studies were conducted in high TB burden countries. Although they are all the studies with small sample size, there is no significant difference between high TB burden countries and low TB burden countries in the differential diagnosis efficacy of HBHA-IGRA through statistical analysis. Perhaps because there are only a low number of studies in high TB burden countries, the pooled results are not as reliable as the results from the countries in low TB burden. Therefore, the conclusion has been softened, the related revised portion is marked in the paper.

5.3. Line 261: “the application of the HBHA-IGRA is restricted by the immune status” Here again, there are too few studies to draw such a conclusion. Studies on the performance of the HBHA-IGRA to detect LTBI (and not for differential diagnosis between ATB and LTBI) among patients under hemodialysis have shown a high diagnostic accuracy of the HBHA-IGRA test which was more sensitive than the TST and the QFT (Dessein R et al PLoS One 2013).

Response: Indeed, the expression “the application of the HBHA-IGRA is restricted by the immune status” here is one-sided and lacks evidence. Therefore, we further checked the relevant literature and revised this part to make the expression more rigorous. The expression “the application of the HBHA-IGRA still need to be further clarified” may be more appropriate here.

5.4. Lines 262-264. If the HBHA-IGRA tests still has to be standardized for a commercial application, it should however be acknowledged that it is the only IGRA test providing actually such a good discrimination between ATB and LTBI. It is therefore now quite urgent that commercial companies became interested to introduce this test in the commercially available portfolio of standardized IGRA.

Response: We are pleased with your comments in the Conclusion. We have revised the relevant expressions in this part, hoping to better express our propose: make HBHA-IGRA get more attention from commercial companies and can be applied to commercial IGRA tests quickly.

Kind regards,

Yueyun Ma

Clinical Laboratory

Air Force Medical Centre

30 Fucheng Road, Beijing, China 100142

---

## [Decision Letter · Decision Letter 1]

10 Jun 2021

PONE-D-21-06709R1

Mycobacterial heparin-binding hemagglutinin (HBHA)-induced interferon-γ release assay (IGRA) for discrimination of latent and active tuberculosis: a systematic review and meta-analysis

PLOS ONE

Dear Dr. Ma,

Thank you for submitting your revised manuscript to PLOS ONE. After careful consideration, we feel that it has merit but still does not fully meet PLOS ONE’s publication criteria as it currently stands. Therefore, we invite you to submit a further revised version of the manuscript that addresses the remaining minor points raised during the review process, before we can proceed for acceptance.

**Also, as reviewer #3 pointed out, I strongly encourage you to have the manuscript edited by a native English speaker.**

We look forward to receiving your revised manuscript.

Kind regards,

Olivier Neyrolles

Academic Editor

PLOS ONE

Journal Requirements:

Reviewers' comments:

Reviewer's Responses to Questions

**Comments to the Author**

1. If the authors have adequately addressed your comments raised in a previous round of review and you feel that this manuscript is now acceptable for publication, you may indicate that here to bypass the “Comments to the Author” section, enter your conflict of interest statement in the “Confidential to Editor” section, and submit your "Accept" recommendation.

Reviewer #1: All comments have been addressed

Reviewer #2: All comments have been addressed

Reviewer #3: (No Response)

2. Is the manuscript technically sound, and do the data support the conclusions?

Reviewer #1: (No Response)

Reviewer #2: Yes

Reviewer #3: Yes

3. Has the statistical analysis been performed appropriately and rigorously? 

Reviewer #1: (No Response)

Reviewer #2: Yes

Reviewer #3: Yes

4. Have the authors made all data underlying the findings in their manuscript fully available?

Reviewer #1: (No Response)

Reviewer #2: Yes

Reviewer #3: Yes

5. Is the manuscript presented in an intelligible fashion and written in standard English?

Reviewer #1: (No Response)

Reviewer #2: Yes

Reviewer #3: No

6. Review Comments to the Author

Reviewer #1: (No Response)

Reviewer #2: please, in table 1, in the column indicating TYPE OF TB, regarding the papers by:

1. Delogu, Plos One 2011

2; Chiacchio, PloS One, 2017:

please indicate that the type of TB was PULMONARY TB in 100% of the TB patients evaluated, as stated in the matherial and method sections where it is written that TB was microbiologically diagnosed on sputum.

Moreover, in the paper by Sali et al, you may see in Table 1, that extrapumonary TB is in 5 over 19 (26%)

Reviewer #3: The authors appropriately answer to my comments and I only have a few remaining comments.

1. Abstract

a. Results – line 40: why did you replace “did” by “”may”? I think “did “ was more appropriate

b. Conclusion. I suggest to delete the last sentence that was added. It is unnecessary and does not correspond to the main message of your meta-analysis. You clearly show that the HBHA-IGRA is very robust and minimally influenced by various technical differences between the studies. Therefore it is not appropriate to conclude that large and high quality studies are further needed. What is necessary now as mentioned at the end of you discussion, is to arise the interest of a commercial company.

2. Discussion

a. I still have a concern concerning the interpretation of the results obtained in studies included in your meta-analysis and comprising HIV-infected patients. You conclude that low responses in these patients may be due to the low CD4+ T cell number in these patients (lines 259-262). However, Wyndham-Thomas et al who were in 2015 the first to evaluate the performance of the HBHA-IGRA in HIV-infected people living in a low TB incidence country, reported that the HBHA-IGRA was more sensitive that both the TST and QFT test to identify potentially Mtb infected people (3 subjects with an isolated HBHA-IGRA had a high Mtb exposure risk). In addition, they showed that the 3 HIV-infected patients with a positive HBHA-IFN-g response had very high responses contrasting to what is reported for non-HIV infected subjects. This observation does not sustain the hypothesis that low CD4+ T cell number account for the low sensitivity of the HBHA IGRA often reported in HIV-infected subjects.

b. Lines 290-291: variability of the cut-off between different studies may also be due to different demographic characteristics of the populations included in the studies.

3. Conclusion

a. Lines 323-324…I suggest “…combination of the results of the HBHA-IGRA with those from other IGRAs (ESAT-6 and CFP-10-based) may allow optimal stratification of Mtb infected patients in different groups with variable risks of reactivation of the infection. Currently, the HBHA-IGRA is the only…..”

b. line 330-331. I do not understand your statement “the procedure of this test needs to be further standardized and optimized” as you clearly showed that technical differences had no impact on the diagnostic performance.

I suggest to modify this sentence as follows: “To commercialize the HBHA-based IGRA to efficiently distinguish LTBI from ATB, it is urgent to rise the interest of commercial companies to this test in order to provide kits with well defined technical conditions and cut off”

4. Table 3

Please correct the subgroups within active TB. I guess the 2nd group is “Clinically confirmed TB”

In the footnote of the table, Mycobacterium smegmatis should be written in italic.

I previously suggest the authors to cite a paper published by Smits K in 2015 and the authors did not find the reference. There was indeed a mistake and I apologize for this. The exact ref is J. Clin. Cell. Immunol. 2015; 6:4 at http://dx.doi.org/10.4172/2155-9899. 1000341

Finally, I strongly suggest the authors to have a final re-lecture and correction by a native English speaking people as the English language should really be improved mostly, but not exclusively, for the modified sentences in the revised version of the manuscript.

7. PLOS authors have the option to publish the peer review history of their article (what does this mean?). If published, this will include your full peer review and any attached files.

Reviewer #1: No

Reviewer #2: No

Reviewer #3: **Yes: **Françoise Mascart

---

## [Author Response · Author response to Decision Letter 1]

20 Jun 2021

Dear Dr. Olivier Neyrolles and Reviewers:

Thank you for your and the reviewers’ comments concerning our manuscript entitled “Mycobacterial heparin-binding hemagglutinin (HBHA)-induced interferon-γ release assay (IGRA) for discrimination of latent and active tuberculosis: a systematic review and meta-analysis” (Manuscript Number: PONE-D-21-06709R1). Your comments and those of the reviewers were highly insightful and enabled us to greatly improve the quality of our manuscript. We have studied comments carefully and have made corrections which we hope meet with approval. The revised portion are marked in the Revised Manuscript with Track Changes. The responds to the reviewer’s comments are as follows:

Reviewer #1: (No Comments)

Reviewer #2: 

Comments for the author

please, in table 1, in the column indicating TYPE OF TB, regarding the papers by:

1. Delogu, Plos One 2011

2; Chiacchio, PloS One, 2017:

please indicate that the type of TB was PULMONRY TB in 100% of the TB patients evaluated, as stated in the material and method sections where it is written that TB was microbiologically diagnosed on sputum. 

Moreover, in the paper by Sali et al, you may see in Table 1, that extrapumonary TB is in 5 over 19 (26%)

Response: Thank you very much for your reminder of the omissions in our paper. The related revised portions are marked in the paper.

Reviewer #3: The authors appropriately answer to my comments and I only have a few remaining comments.

Comments for the author

1. Abstract:

a. Results – line 40: why did you replace “did” by “may”? I think “did” was more appropriate

Response: Thank you for your advice. The related revised portion is marked in the paper.

b. Conclusion. I suggest to delete the last sentence that was added. It is unnecessary and does not correspond to the main message of your meta-analysis. You clearly show that the HBHA-IGRA is very robust and minimally influenced by various technical differences between the studies. Therefore, it is not appropriate to conclude that large and high quality studies are further needed. What is necessary now as mentioned at the end of your discussion, is to arise the interest of a commercial company.

Response: Thanks. Your comment is right. The last sentence here may cause misunderstanding and confusion. Therefore, we delete this sentence according to your suggestion.

2. Discussion:

a. I still have a concern concerning the interpretation of the results obtained in studies included in your meta-analysis and comprising HIV-infected patients. You conclude that low responses in these patients may be due to the low CD4+ T cell number in these patients (lines 259-262). However, Wyndham-Thomas et al who were in 2015 the first to evaluate the performance of the HBHA-IGRA in HIV-infected people living in a low TB incidence country, reported that the HBHA-IGRA was more sensitive that both the TST and QFT test to identify potentially Mtb infected people (3 subjects with an isolated HBHA-IGRA had a high Mtb exposure risk). In addition, they showed that the 3 HIV-infected patients with a positive HBHA-IFN-g response had very high responses contrasting to what is reported for non-HIV infected subjects. This observation does not sustain the hypothesis that low CD4+ T cell number account for the low sensitivity of the HBHA IGRA often reported in HIV-infected subjects.

Response: Considering your suggestion, we have substantially elucidated both the outcome of subgroup analysis in HIV-infected people and the observation in the study of Wyndham-Thomas et al in 2015 in the discussion. we consider that the reasons for the low sensitivity of the HBHA-IGRA in HIV-infected patients are complex, and suggest that the mechanism of HIV infection affecting HBHA-induced IFN-γ release still needs further investigation. The specific mechanism is not the main focus of the discussion in this paper.

b. Lines 290-291: variability of the cut-off between different studies may also be due to different demographic characteristics of the populations included in the studies.

Response: We support your suggestion. As you suggested, the related discussion has been added in the limitation.

3. Conclusion

a. Lines 323-324…I suggest “…combination of the results of the HBHA-IGRA with those from other IGRAs (ESAT-6 and CFP-10-based) may allow optimal stratification of Mtb infected patients in different groups with variable risks of reactivation of the infection. Currently, the HBHA-IGRA is the only….”

Response: Thank you for your suggestion. The related revised portion is marked in the paper.

b. line 330-331. I do not understand your statement “the procedure of this test needs to be further standardized and optimized” as you clearly showed that technical differences had no impact on the diagnostic performance.

I suggest to modify this sentence as follows: “To commercialize the HBHA-based IGRA to efficiently distinguish LTBI from ATB, it is urgent to rise the interest of commercial companies to this test in order to provide kits with well defined technical conditions and cut off”

Response: Thank you for your suggestion. We have re-written this part according to your suggestion. The related revised portion is marked in the paper.

4. Table 3

Please correct the subgroups within active TB. I guess the 2nd group is “Clinically confirmed TB”

In the footnote of the table, Mycobacterium smegmatis should be written in italic.

Response: We are sorry for our incorrect writing. The related revised portions are marked in the paper.

I previously suggest the authors to cite a paper published by Smits K in 2015 and the authors did not find the reference. There was indeed a mistake and I apologize for this. The exact ref is J. Clin. Cell. Immunol. 2015; 6:4 at http://dx.doi.org/10.4172/2155-9899. 1000341

Response: Thank you very much for providing the specific information of the reference, the reference has been cited in this paper.

Finally, I strongly suggest the authors to have a final re-lecture and correction by a native English speaking people as the English language should really be improved mostly, but not exclusively, for the modified sentences in the revised version of the manuscript.

Response: Thanks for pointing out our language problem. We employed an English-language editing service, MedSci, to polish our wording. Certification is attached. 

Kind regards,

First Author: Jinhua Tang

Corresponding author: Yueyun Ma 

E-mail addresses: mayueyun2020@163.com

Clinical Laboratory

Air Force Medical Centre

30 Fucheng Road, Beijing, China 100142

---

## [Decision Letter · Decision Letter 2]

30 Jun 2021

Mycobacterial heparin-binding hemagglutinin (HBHA)-induced interferon-γ release assay (IGRA) for discrimination of latent and active tuberculosis: a systematic review and meta-analysis

PONE-D-21-06709R2

Dear Dr. Ma,

We’re pleased to inform you that your manuscript has been judged scientifically suitable for publication and will be formally accepted for publication once it meets all outstanding technical requirements.

Kind regards,

Olivier Neyrolles

Section Editor

PLOS ONE

Additional Editor Comments (optional):

**During the proofreading process, please proceed to the following modifications:**

**Line 274 « nHBHA frequently responded to the T cells from the LTBI subjects …..should be replaced by “T cells from LTBI subjects who showed …..frequently responded to nHBHA”**

**Line 296  “the TST results are often positive than the IGRAs” should be replaced by “the TST results are more often positive than the IGRAs”**

Reviewers' comments:

Reviewer's Responses to Questions

**Comments to the Author**

1. If the authors have adequately addressed your comments raised in a previous round of review and you feel that this manuscript is now acceptable for publication, you may indicate that here to bypass the “Comments to the Author” section, enter your conflict of interest statement in the “Confidential to Editor” section, and submit your "Accept" recommendation.

Reviewer #3: All comments have been addressed

2. Is the manuscript technically sound, and do the data support the conclusions?

Reviewer #3: Yes

3. Has the statistical analysis been performed appropriately and rigorously? 

Reviewer #3: Yes

4. Have the authors made all data underlying the findings in their manuscript fully available?

Reviewer #3: Yes

5. Is the manuscript presented in an intelligible fashion and written in standard English?

Reviewer #3: Yes

6. Review Comments to the Author

Reviewer #3: (No Response)

7. PLOS authors have the option to publish the peer review history of their article (what does this mean?). If published, this will include your full peer review and any attached files.

Reviewer #3: **Yes: **Françoise Mascart

---

## [Editor Report · Acceptance letter]

7 Jul 2021

PONE-D-21-06709R2 

Mycobacterial heparin-binding hemagglutinin (HBHA)-induced interferon-γ release assay (IGRA) for discrimination of latent and active tuberculosis: a systematic review and meta-analysis 

Dear Dr. Ma:

I'm pleased to inform you that your manuscript has been deemed suitable for publication in PLOS ONE. Congratulations! Your manuscript is now with our production department. 

Kind regards, 

on behalf of

Dr. Olivier Neyrolles 

Section Editor

PLOS ONE